# Crop Phenology Modelling Using Proximal and Satellite Sensor Data

Anne Gobin [1,2,*] , Abdoul-Hamid Mohamed Sallah [3], Yannick Curnel [4], Cindy Delvoye [5], Marie Weiss [6] , Joost Wellens [3] , Isabelle Piccard [2] , Viviane Planchon [4], Bernard Tychon [3], Jean-Pierre Goffart [4] and Pierre Defourny [5]

1   Department of Earth and Environmental Sciences, Faculty of Bioscience Engineering, Katholieke Universiteit Leuven, 3001 Leuven, Belgium

2   Remote Sensing Unit, Flemish Institute of Technological Research (VITO), 2400 Mol, Belgium; isabelle.piccard@vito.be

3   SPHERES Research Unit, University of Liège, 6700 Arlon, Belgium; ahamidms@uliege.be (A.-H.M.S.); joost.wellens@uliege.be (J.W.); bernard.tychon@uliege.be (B.T.)

4   Centre Wallon de Recherches Agronomiques, CRAW, 5030 Gembloux, Belgium; y.curnel@cra.wallonie.be (Y.C.); v.planchon@cra.wallonie.be (V.P.); j.goffart@cra.wallonie.be (J.-P.G.)

5   Earth and Life Institute, Université Catholique de Louvain, 1348 Louvain-la-Neuve, Belgium; cindy.delvoye@uclouvain.be (C.D.); pierre.defourny@uclouvain.be (P.D.)

6   EMMAH, Institut National de Recherche pour l'Agriculture, l'alimentation et l'Environnement (INRAE), 84000 Avignon, France; marie.weiss@inrae.fr

*   Correspondence: anne.gobin@kuleuven.be

**Abstract:** Understanding crop phenology is crucial for predicting crop yields and identifying potential risks to food security. The objective was to investigate the effectiveness of satellite sensor data, compared to field observations and proximal sensing, in detecting crop phenological stages. Time series data from 122 winter wheat, 99 silage maize, and 77 late potato fields were analyzed during 2015–2017. The spectral signals derived from Digital Hemispherical Photographs (DHP), Disaster Monitoring Constellation (DMC), and Sentinel-2 (S2) were crop-specific and sensor-independent. Models fitted to sensor-derived fAPAR (fraction of absorbed photosynthetically active radiation) demonstrated a higher goodness of fit as compared to fCover (fraction of vegetation cover), with the best model fits obtained for maize, followed by wheat and potato. S2-derived fAPAR showed decreasing variability as the growing season progressed. The use of a double sigmoid model fit allowed defining inflection points corresponding to stem elongation (upward sigmoid) and senescence (downward sigmoid), while the upward endpoint corresponded to canopy closure and the maximum values to flowering and fruit development. Furthermore, increasing the frequency of sensor revisits is beneficial for detecting short-duration crop phenological stages. The results have implications for data assimilation to improve crop yield forecasting and agri-environmental modeling.

**Keywords:** precision agriculture; Copernicus Sentinel-2 (S2); disaster monitoring constellation (DMC); digital agriculture; remote sensing; arable cop

## 1. Introduction

An important aspect of estimating crop yield and production concerns the stages of crop growth and development, known as crop phenology. Crop phenology affects almost all aspects of the agri-environment, including yield formation, water and carbon cycling, agro-ecosystem services, and food webs. Changes in crop phenological processes such as flowering are among the most sensitive biological responses to climate change [1], so crop phenological development stages allow comparisons of impacts between regions and seasons. Globally, many phenological stages, including the start and end of the growing season, are occurring earlier in the spring and later in the autumn [2–4]. However, not

all species and regions are affected at the same rate, leading to mismatches and making phenology a leading indicator of climate change impacts [5,6].

The occurrence of meteorological events, such as frost or drought, during the crop life cycle or at specific phenological stages explains yield variation in common crops [7–10]. Crop phenology and canopy development help improve the effectiveness of management practices and the accuracy of decision support systems [11,12]. Farmers need to know the timing of crop development to decide when to apply fertilizers, pesticides, or remedial measures and to implement certain practices, such as irrigation, in a more sustainable way. Crop phenology models are the basis of all crop models, and crop emergence, flowering, and maturity are typically simulated from weather variables [13–15]. Comparisons with measured crop data consist of observations and intensive within-season time series information on soil water content, crop biomass, and yield [15–17].

The accuracy of arable crop models largely depends on a reliable estimation of the crop phenological stages to which the processes of crop performance and yield formation are tuned. Understanding crop phenology is crucial for predicting crop yields and identifying potential climate risks to food security [14,18,19]. The productivity of a crop depends on its development and ability to intercept incident radiation, which is a function of the leaf area and structure of the crop [20]. All 27 crop models included in [16] simulate wheat (*Triticum aestivum*) phenology based on planting date ($n = 27$) and temperature summation approaches ($n = 27$); some models also consider the effects of daylength and vernalization ($n = 21$) or water/nutrient stress ($n = 6$). Based on 23 participating models, the maize (*Zea mays*) crop model intercomparison [17] shows that days after sowing and temperature ($n = 23$) are the most important factors used for deriving phenological stages, followed by day length ($n = 15$) and nutrient or water stress ($n = 5$). An intercomparison review of 33 potato (*Solanum tuberosum*) models showed that most research has been carried out in relation to soil-water-nitrogen dynamics, with the phenological compartment being driven by days after planting ($n = 33$), or temperature ($n = 30$) in combination with daylength ($n = 13$) [21].

Vegetation phenology can be monitored with vegetation indices (VI) derived from optical remote sensing [22,23]. The need for monitoring crop phenology at the farm-to-regional scale has become apparent to, inter alia, clarify crop responses to seasonal variation, elucidate effective farm management practices, and improve crop performance prediction and yield forecasting. As the growing season progresses, the crop's greenness increases, leaving a distinct and detectable spectral signature that changes with different stages of development [24,25]. Methods to detect phenological stages range from the detection of single phenological events based on NDVI to complex mathematical modeling methods using VI time series analysis and sensor fusion methods to infer crop phenological development [26–29]. Despite the disadvantage of requiring a priori information such as the number of cropping seasons during the year [30] and planting/sowing dates [31], model fitting offers the possibility to mathematically characterize specific transitions in the crop phenological cycle [32,33]. Most methods determine crop growth after the growing season using time-series smoothing techniques such as splines or neural networks [34,35]. Similar smoothing and machine learning methods have been developed to evaluate yield estimates at farm-to-regional scales [36,37], compare different growing seasons [38], and support decisions on farming practices [11]. However, model fitting would enable a mathematical characterization of changes in crop growth rate and, therefore, crop phenology during the growing season, providing information on crop performance independent of currently used weather-based crop modeling methods.

Since crop phenology is a leading indicator of climate impacts, crop performance processes, and yield formation, its detection during the growing season may provide data that is independent of current weather-based modeling methods for simulating crop phenological stages. We hypothesized that biophysical variables derived from satellite sensors provide a valid set of measurements for inferring crop phenology. The objectives of this study are therefore to relate in-situ field phenological observations to proximal and

satellite sensor data and investigate how well modeling using sensor data can be used to infer crop phenological stages during the growing season. A unique dataset of field observations was constructed during three growing seasons (2015–2017) for three major arable crops in Belgium, i.e., silage maize, winter wheat, and late potato.

## 2. Materials and Methods

The proximal sensor data consisted of field-based digital hemispherical photographs (DHP) acquired by a digital camera with a fisheye lens, while the satellite sensor data consisted of disaster monitoring constellation-2/Deimos-1 satellite (DMC) images at a 22 m resolution from Deimos Imaging and Copernicus Sentinel-2 (S2) images at a 10 m resolution from the European Space Agency. All sensor data were processed and analyzed at the parcel level following the flow diagram of data processing presented in Figure 1. Parcel boundaries were retrieved from the Land Parcel Information System (LPIS) used to administer farmers' income support through the European Common Agricultural Policy.

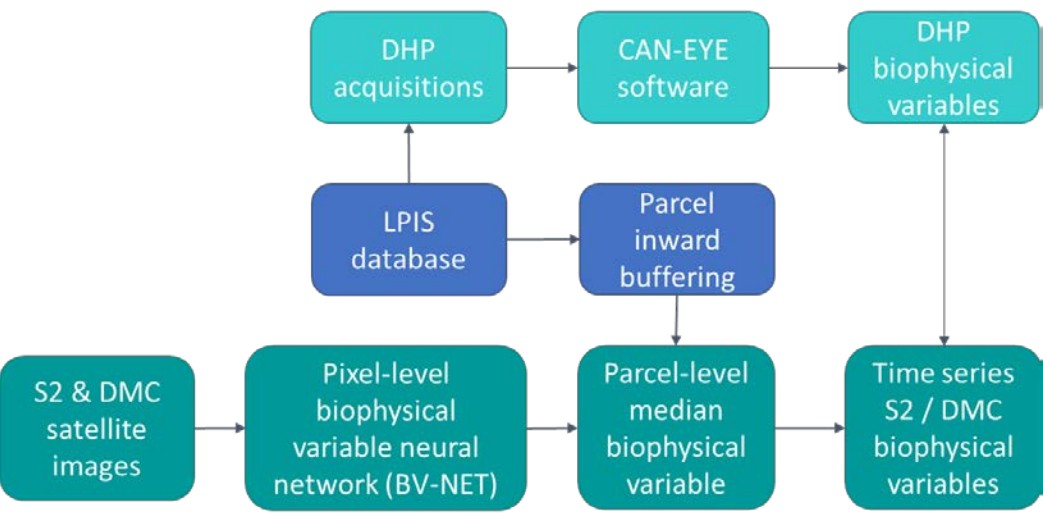

**Figure 1.** Flow diagram of data processing.

### 2.1. Satellite Sensor Observations

A time series of cloud-free ortho-rectified 22 m resolution DMC and 10 m resolution S2 satellite images were used for deriving the biophysical variables fAPAR (fraction of absorbed photosynthetically active radiation) and fCover (vegetation cover fraction) (Table 1). Pre-processing included atmospheric correction [39]; cloud and shadow detection [40] for DMC; [41] for S2; and calculation of the biophysical variables [42].

**Table 1.** Cropping calendar for winter wheat, silage maize, and late potato, and number of fortnightly cloud-free images used during the growing seasons of April–November 2015, 2016, and 2017 in Belgium.

| Crop | Apr. | | May | | Jun. | | Jul. | | Aug. | | Sep. | | Oct. | | Nov. | |
|---|---|---|---|---|---|---|---|---|---|---|---|---|---|---|---|---|
| Winter wheat | | | | | f | f | | h | h | | | | s | s | | |
| Silage maize | | | s | s | e | | | | f | f | | h | h | | | |
| Late potato | p | p | e | e | | | t | t | | | | h | h | | | |
| 2015 DMC (22 m) | | | | | | 3 | 2 | | 2 | 2 | 1 | | | | | |
| 2016 DMC (22 m) | | | | | 3 | 2 | 2 | 4 | 2 | 4 | 4 | | | | | |
| 2015 Sentinel-2A (10 m) | | | | | | | 1 | 2 | 4 | 4 | 3 | 1 | | | | 1 |
| 2016 Sentinel-2A (10 m) | 5 | 4 | 5 | 4 | 2 | 1 | 4 | 5 | 3 | 4 | 4 | 5 | 4 | 5 | 4 | 4 |
| 2017 Sentinel-2A&B (10 m) | 3 | 6 | 3 | 4 | 3 | 5 | 7 | 8 | 5 | 8 | 5 | 9 | 5 | 9 | 5 | 6 |

s: sowing, p: planting (brown); e: emergence (green); f: flowering, t: tuber setting (light green); h: harvesting (dark green). The number of images available is colored from red to dark green.

The derivation of fAPAR and fCover products from Sentinel-2 and DMC was achieved by using the BV-NET (biophysical variable neural network), initially developed for medium-resolution sensors [43], adapted to higher spatial resolution sensors such as SPOT [44] and Landsat-8 [45], and based on neural networks trained on a synthetic dataset of around 50,000 simulations using the PROSAIL model [46]. The distribution of the variables was designed with an experimental plan, and constraints were applied to the co-distribution of the leaf area index (LAI), soil brightness, chlorophyll, and average leaf angle [47]. For Sentinel-2, neural networks derived from the same algorithm are available in the SNAP (Sentinel Application Platform) toolbox [48]. This algorithm showed good performances for Sentinel-2 against in situ measurements, with an RMSE of 0.32 for LAI for five different crops, including wheat and maize [48] and an RMSE of 0.1 for sunflower for fAPAR estimated from Landsat. Similar results were obtained using Formosat data [44].

Each neural network, i.e., one for each sensor and biophysical variable, was calibrated using the same set of PROSAIL inputs: canopy architecture, leaf constituents, and soil background. However, the simulations were adapted to each sensor by considering the spectral bands and orbital characteristics of the particular sensor. To achieve a good temporal and spatial consistency between the two sensors and for inter-comparison reasons, we considered only similar spectral bands between DMC and Sentinel-2, i.e., green, red, and near-infrared bands (Figure 2). Furthermore, this selection allowed us to use the best Sentinel-2 spatial resolution (10 m) and the best accuracy when applying for registration between the two sensors. The inputs of the different neural networks are the reflectance in three spectral bands and the geometry of acquisition, which is determined by the cosine of the sun, the view zenith angle, and the relative azimuth angle.

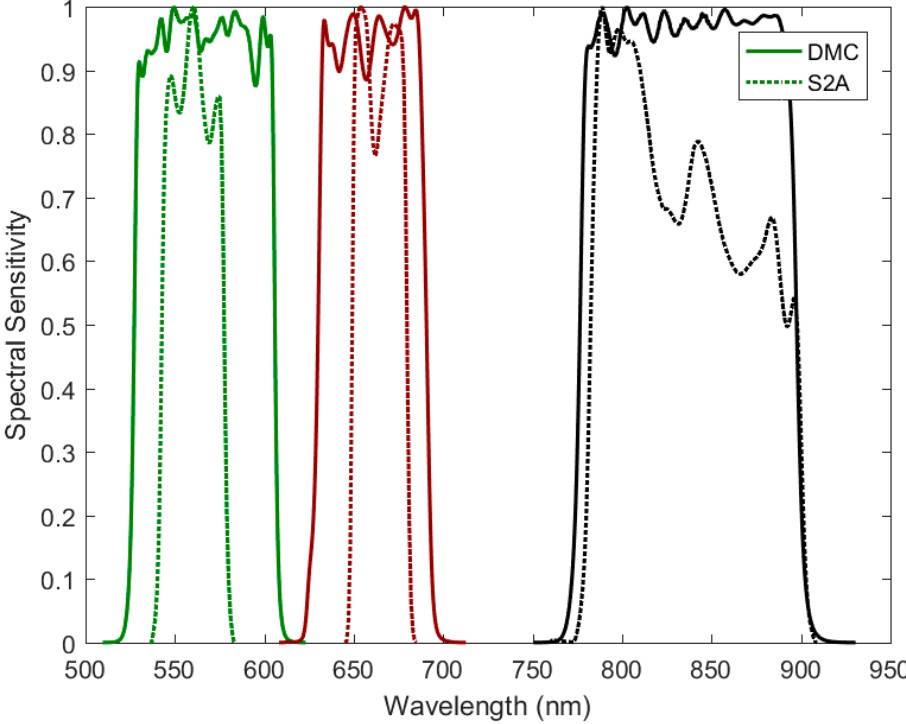

**Figure 2.** Spectral Sensitivity of DMC-Deimos and Sentinel-2A in three spectral bands (green, red, and near-infrared) as inputs to the neural network.

During 2015–2017, a total of 122 fields of winter wheat, 99 maize fields, and 77 potato fields were monitored across Belgium (Table 2). The fields were located in the agroecological regions of Belgium, where the crops occurred mostly (Figure 3). An inward buffer was imposed such that only pure pixels were extracted per field (Figure 4), whereafter the median value per bio-physical variable was retained.

**Table 2.** Arable fields monitored. In brackets are the fields monitored with Digital Hemispheric Photography (DHP).

| Crop | 2015 | 2016 | 2017 | Total |
|------|------|------|------|-------|
| Silage Maize | 10 (10) | 20 (09) | 69 (15) | 99 (34) |
| Late Potato | 20 (01) | 29 (05) | 28 (06) | 77 (12) |
| Winter wheat | 37 (19) | 22 (09) | 63 (10) | 122 (38) |

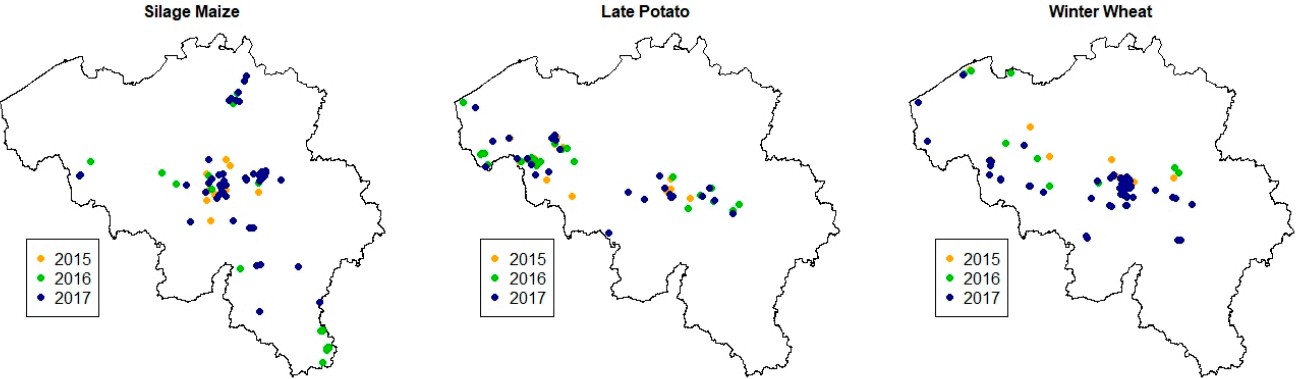

**Figure 3.** Location of arable fields in Belgium for 2015, 2016, and 2017 surveys of silage maize, late potato, and winter wheat.

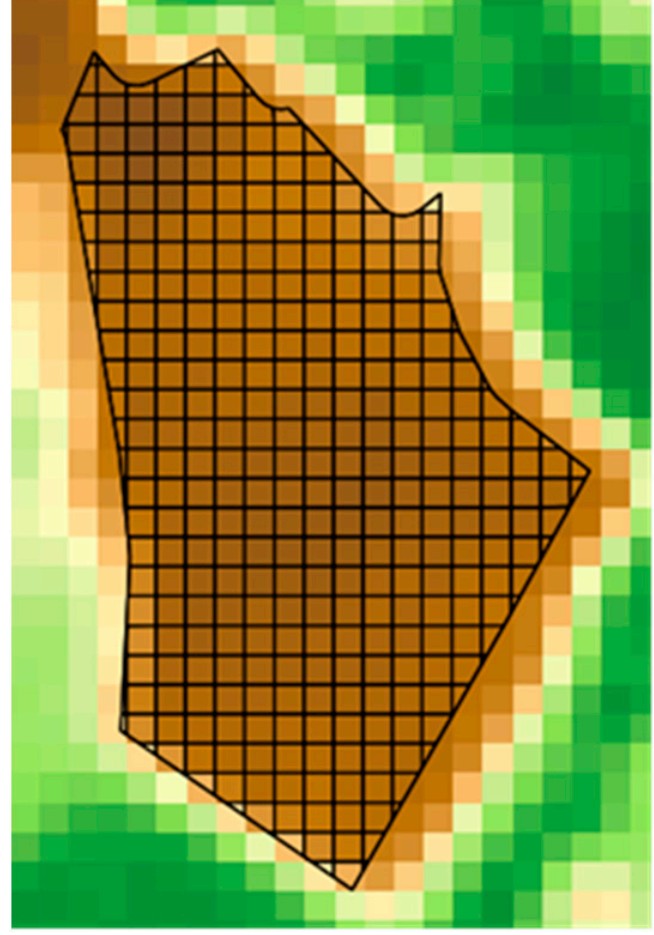

**Figure 4.** Extraction of pixel values contributing to the median satellite-derived biophysical variables at the field scale. Example of DMC-derived fAPAR after inward buffering.

### 2.2. Digital Hemispherical Photography and Ground Observations

Between March and September, all fields were regularly visited to record the phenological stage according to the BBCH-scale. BBCH stands for Biologische Bundesanstalt für Forst und Landwirtschaft, Bundessortenamt für CHemische Industrie, and the description should apply to at least 50% of the individual plants in a field [49]. Pictures were taken with Digital Hemispherical Photography (DHP) in a subset of fields (Table 2; Figure 5). DHP acquisitions were obtained along 2 transects of 15 m, with 5 images taken per transect [50]. The reflectances were established for an area of 20 m resolution centered along the transects. The camera and fish-eye lens was calibrated, and the physical position of the optical center and the projection function were used to undistort the images. The plant area was estimated from the identification of background (soil or sky) and plant (green) pixels and the established geometric considerations. The images were processed using the CAN-EYE software to extract the fraction of absorbed photosynthetically active radiation (fAPAR) and the vegetation cover fraction (fCover) [51]. Commensurate with the BBCH descriptions, median values per field were calculated.

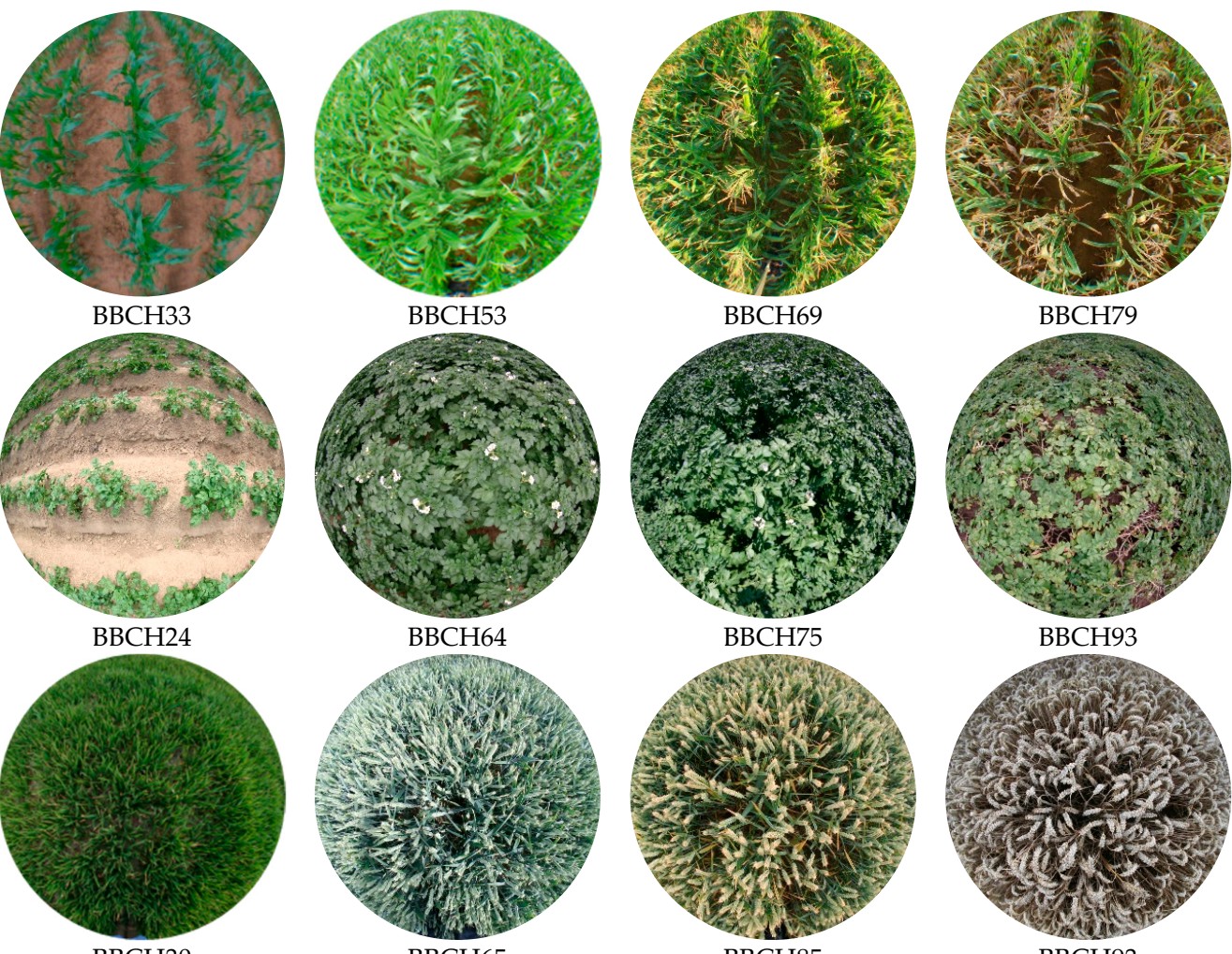

**Figure 5.** Examples of Digital Hemispherical Photos (DHP) and BBCH code for maize (top); potato (middle); and, winter wheat (bottom). An overview of BBCH codes for the different crops is provided in [49].

### 2.3. Statistical Modelling Methods

The dates of image acquisition were transformed to days after planting (DAP) based on the sowing and planting dates recorded for the different winter wheat, potato, and

maize parcels. The coefficient of variation (CV) was used to represent the variation in the biophysical variables fAPAR and fCover between crops, fields, and dates during the cropping season. Using a two-way ANOVA [52], we tested the hypothesis that there were statistically significant relationships between the biophysical variables fAPAR and fCover during the crop growing season and crop type, field location, days after planting, and their factorial interactions.

Subsequent model fitting of all DHP-, DMC-, and Sentinel-2-derived fAPAR and fCover against DAP and/or against the day of the year (DOY) was done in R [52] using maximum likelihood estimation (MLE) for establishing model parameters. Different models were compared for different sensors, i.e., cubic splines, single sigmoid models (Equation (1)), and double sigmoid models, and discussed in terms of goodness-of-fit and practical application. Double-sigmoid models were fitted simultaneously, whereby the decay was defined for *f(x) > f(x)max*.

$$f(x) = \frac{a}{\left[1 + e^{-b(x-c)}\right]} \; for \; f(x) \leq f(x)max \tag{1}$$

where $x$ was expressed in DAP or DOY, $f(x)$ represented the biophysical parameter fAPAR or fCover at time $x$; and $a$, $b$, and $c$ were fitting parameters. The best model fit was based on the lowest estimators of prediction error using the Akaike information criterion (AIC, [53]). The goodness-of-fit statistics between observed and estimated values were mean average error (MAE), root mean square error (RMSE), coefficient of determination ($R^2$), and model agreement (d) [54]. For both the upward and downward sigmoid models, the inflection points, the endpoint (saturation), and the maximum fAPAR and fCover values were derived and subsequently related to the BBCH code.

## 3. Results

### 3.1. Digital Hemispherical Photography and Ground Observations

The crop phenological stages through which maize, potato, and wheat crops passed and their durations were expressed in time after planting. Despite the categorical scale and crisp delineation of BBCH codes identified in the field, the crop phenological stages showed a clear progression of growth stages toward maturity and harvest (Figure 6). The BBCH codes were related to days after planting/sowing (DAP).

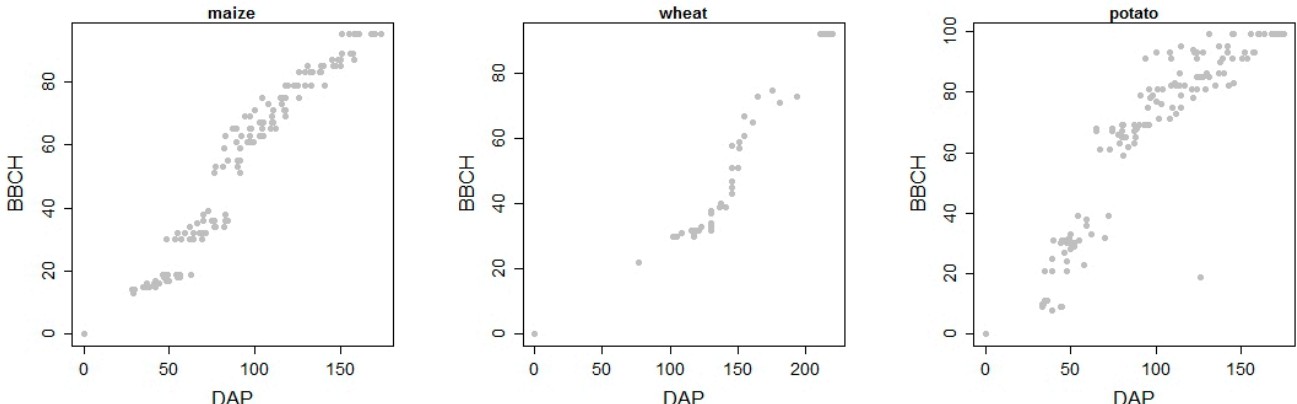

**Figure 6.** Observed BBCH codes versus days after planting/sowing (DAP) for maize, winter wheat, and potato for the period 2015–2017.

For maize, leaf development (BBCH: 10–19) corresponded to 28–63 days after sowing, stem elongation (BBCH: 30–39) to 48–84 DAP, heading or inflorescence emerging (BBCH: 51–59) to 76–91 DAP, flowering/anthesis (BBCH: 60–69) to 83–118 DAP, grain development (BBCH: 70–79) to 100–141 DAP, ripening (BBCH: 80–89) to 126–158 DAP, and senescence (BBCH: 90–95) to above 160 DAP.

For winter wheat, emergence and early leaf development took place before winter. Leaf development was re-commenced after winter and was followed by a period of stem elongation between 102–141 days after sowing. The booting stage (138–146 DAP) preceded heading (146–151 DAP) and anthesis (155–161 DAP). Maturity stages comprise grain development and ripening between 170–194 DAP, followed by senescence. Harvest occurred around 200–220 DAP.

Potato BBCH stages were distinctly different from cereal phenology. Sprouting and germination (BBCH: 0–10) took one month on average and were closely followed by leaf development and crop emergence (BBCH: 10–20). Formation of side shoots (BBCH: 20–30) stretched between 35–52 DAP, while main stem elongation (BBCH: 30–40) lasted for 45–72 DAP. The onset of tuber formation (BBCH: 40–50) coincided with inflorescence emerging (BBCH: 50–60) and flowering (BBCH: 60–70) during the period 65–96 DAP. The duration of the above-ground development of fruit (BBCH: 70–80) was at 91–115 DAP, whereas seed development (BBCH: 80–90) was at 96–138 DAP. Senescence covered a period of 122–175 DAP.

Digital hemispherical photography (DHP) enabled in situ ground observations, whereby the analysis of DHP-derived fCover and fAPAR provided a good insight into the ability to detect crop phenology (Table 3 and Figure 7) that relates to satellite sensor-derived biophysical variables. Both variables changed during crop growth and development, and their specific characteristics reflected different phenological stages. An excellent goodness-of-fit was obtained for fitted growth models of fAPAR and fCover for individual silage maize fields (Table 3). Compared to silage maize, the goodness-of-fit of the fAPAR and fCover models for individual winter wheat fields was more variable. For individual late potato fields, the growth models on fAPAR and fCover displayed the largest variability. Overall, DHP-derived fAPAR captured the dynamics of phenological development and the retrieval of canopy closure better at the field level than DHP-derived fCover.

**Table 3.** Goodness-of-fit metrics for model fitting per field of DHP-derived fAPAR and fCover.

| Crop | Variable | MAE | RMSE | $R^2$ | d |
|---|---|---|---|---|---|
| Silage Maize | fAPAR | 0.03–0.08 | 0.03–0.09 | 0.75–0.99 | 0.93–0.99 |
| | fCover | 0.01–0.09 | 0.01–0.10 | 0.75–0.99 | 0.93–0.99 |
| Late Potato | fAPAR | 0.00–0.11 | 0.00–0.13 | 0.46–0.99 | 0.81–0.99 |
| | fCover | 0.00–0.14 | 0.00–0.15 | 0.55–0.99 | 0.50–0.99 |
| Winter wheat | fAPAR | 0.00–0.09 | 0.00–0.09 | 0.60–0.99 | 0.53–0.99 |
| | fCover | 0.00–0.18 | 0.00–0.24 | 0.43–0.99 | 0.82–0.99 |

### 3.2. Satellite Observations

The development of biophysical products from satellite sensors enabled the construction of time series per field (Figure 8), provided that cloud contamination was minimal during the season.

Similar findings to DHP were observed for growth models fitted from DMC-derived time series of fCover and fAPAR. In the case of maize, the obtained goodness-of-fit per field was similar for fAPAR (MAE = 0.014–0.031; RMSE = 0.014–0.040; $R^2$ = 0.96–0.99; d = 0.99–1) as for fCover (MAE = 0.019–0.039; RMSE = 0.024–0.048; $R^2$ = 0.95–0.99; d = 0.99–1) (Figure 9). The biophysical variable fCover displayed less uncertainty during the onset of the crop growing season, whereas fAPAR displayed less uncertainty during vegetative development and mature crop growth stages.

The comparatively higher revisit frequency and higher spatial resolution increase the probability and number of pure and cloud-free pixels per field for Sentinel-2 as compared to DMC (Figure 10). Models that suited all fields per crop were difficult to develop since the probability of cloud- or shadow-contaminated values also increased with higher revisit frequencies, adding to the variability between fields and years.

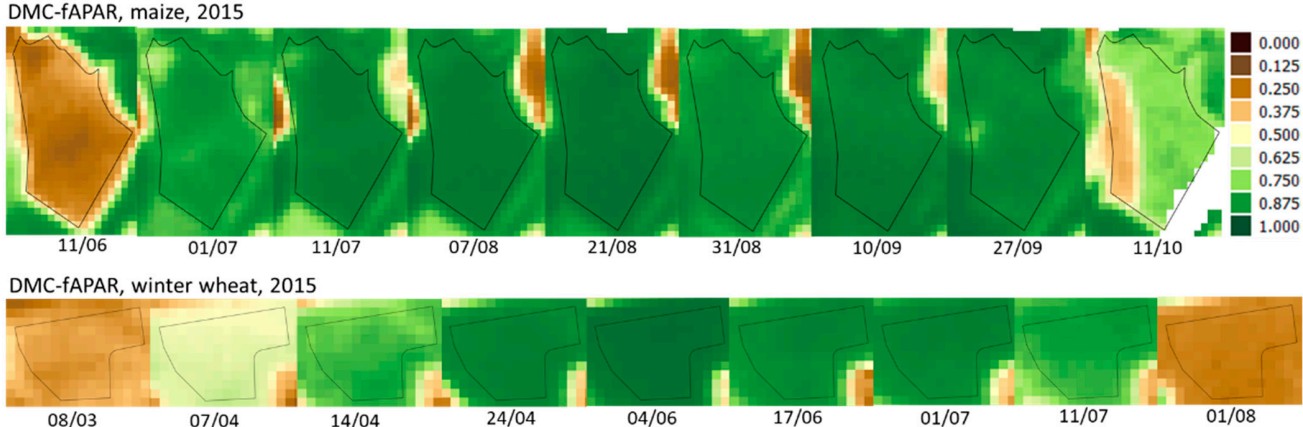

**Figure 7.** DHP-derived fCover and fAPAR versus days after planting/sowing (DAP) for maize, winter wheat, and potato for the period 2015–2017.

DMC-fAPAR, maize, 2015

| 11/06 | 01/07 | 11/07 | 07/08 | 21/08 | 31/08 | 10/09 | 27/09 | 11/10 |

DMC-fAPAR, winter wheat, 2015

| 08/03 | 07/04 | 14/04 | 24/04 | 04/06 | 17/06 | 01/07 | 11/07 | 01/08 |

**Figure 8.** DMC-derived fAPAR progression for maize and winter wheat fields during 2015, including day/month of image acquisition.

Models per field highlighted the location-specific nature of crop development as demonstrated by cubic spline fits for the three different crops (Figure 11). For maize, we chose a time window for the growing period between 1st May and 1st October; for potatoes, between 10 April and 1 October; and, for winter wheat, we considered the growing period after vernalization between 10 February and 1 August. Three knots were defined at 50-day intervals during the growing season, and cubic polynomials were subsequently fitted

through the fAPAR and fCover points. However, the splined models could only be fitted after the cropping season.

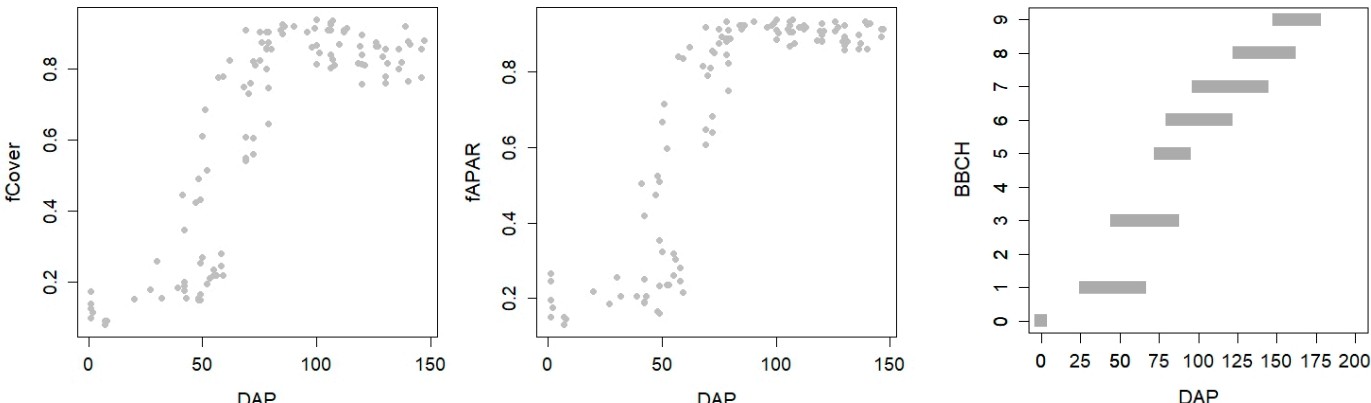

**Figure 9.** DMC-derived fCover and fAPAR versus days after planting/sowing (DAP) for maize fields in 2015 and related BBCH codes versus DAP.

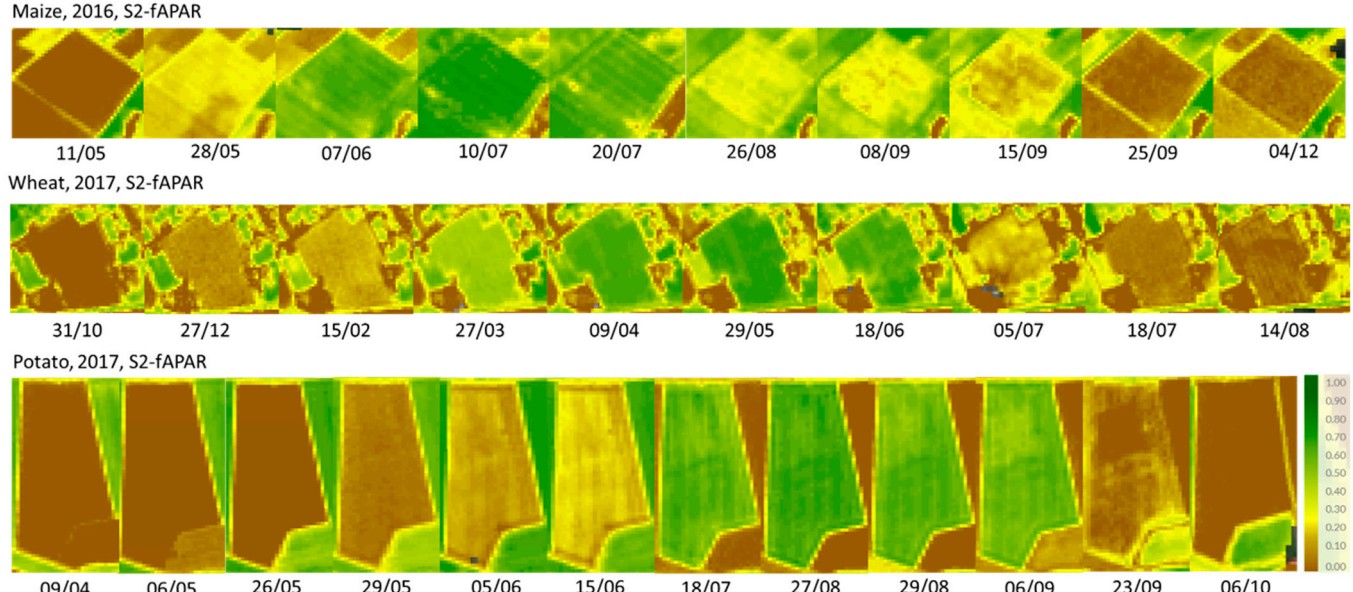

**Figure 10.** Sentinel-2 derived fAPAR progression for maize, winter wheat, and potato fields during 2016 and 2017, including the day/month of image acquisition.

### 3.3. Differences and Similarities between Sensors during the Cropping Season

A comparison between DHP, DMC, and Sentinel-2-derived fCover and fAPAR during the growing season demonstrated similarities between the different sensors, whereby each field is represented by the median value of pixels in the case of satellite sensors or by the median of DHP measurements (Figure 12). The comparison showed a strong relationship between DMC and DHP and between Sentinel-2 and DHP. Overall, the relationship was stronger for fAPAR than for fCover.

An analysis of variance (ANOVA) demonstrated clear effects of crops, fields, dates, and their factorial interactions on fAPAR ($p < 0.001$) and fCover ($p < 0.001$). For maize in 2015, fAPAR (fCover) showed the largest variability during the onset of the season, with the coefficient of variation (cv) dropping exponentially as the season progressed, from 87% (125%) at sowing to 12% (12%) at stem elongation, 3% (3%) at maturity, and 5% (5%) at senescence. For wheat in 2015, the cv in fAPAR (fCover) dropped from 46% (67%) at stem elongation to 9% (11%) at maturity and back up to 21% (25%) at senescence. The variability (cv) in fAPAR (fCover) ranged between 32% and 56% for maize fields and between 36%

(41%) and 65% (78%) for wheat fields; these ranges were overall larger than the variability between dates. For all crops, fAPAR consistently showed less variability for the different crop development stages.

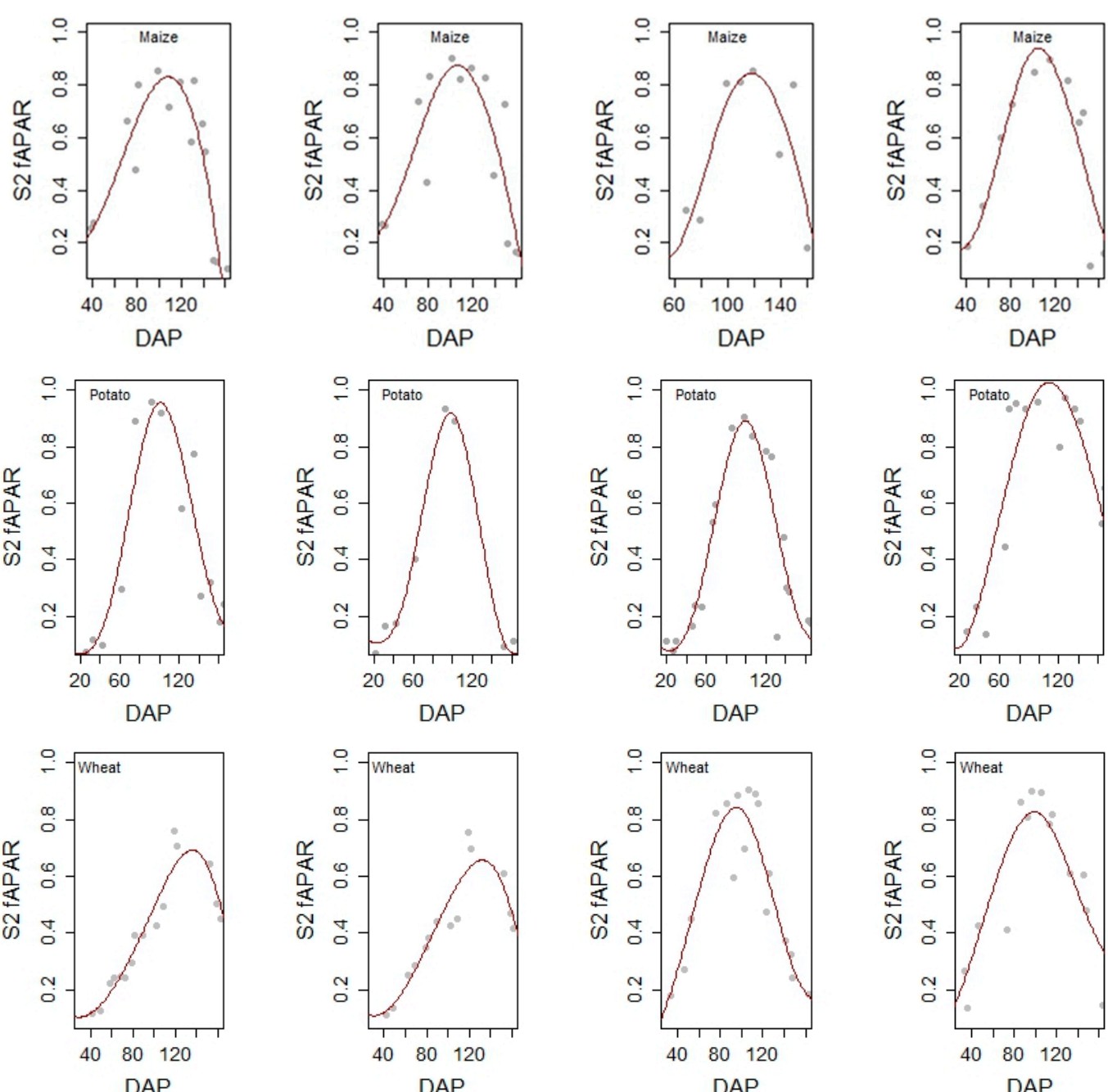

**Figure 11.** S2-derived fAPAR and cubic splines projected on days after planting/sowing (DAP) for selected fields of maize, potato, and winter wheat during the 2017 growing season.

*3.4. Crop Phenology Detection Differences and Similarities between Sensors during the Cropping Season*

Model fitting of crop phenological development concentrated on relating field observations in the BBCH scale to Sentinel-2-derived fAPAR time series, since this biophysical variable provided better results than fCover during the crop growing season. A numerical solution of fitting models through fAPAR time series enabled the mathematical derivation of inflection points and the subsequent detection of changes in green vegetation status and growth rate. The inflection points (Figure 13) represented the times of rapid, peak, and

decay green vegetation development. The fit for above ground crop development was highest for maize, followed by winter wheat and potato, demonstrating the efficiency of Sentinel-2 satellite imagery for crop monitoring.

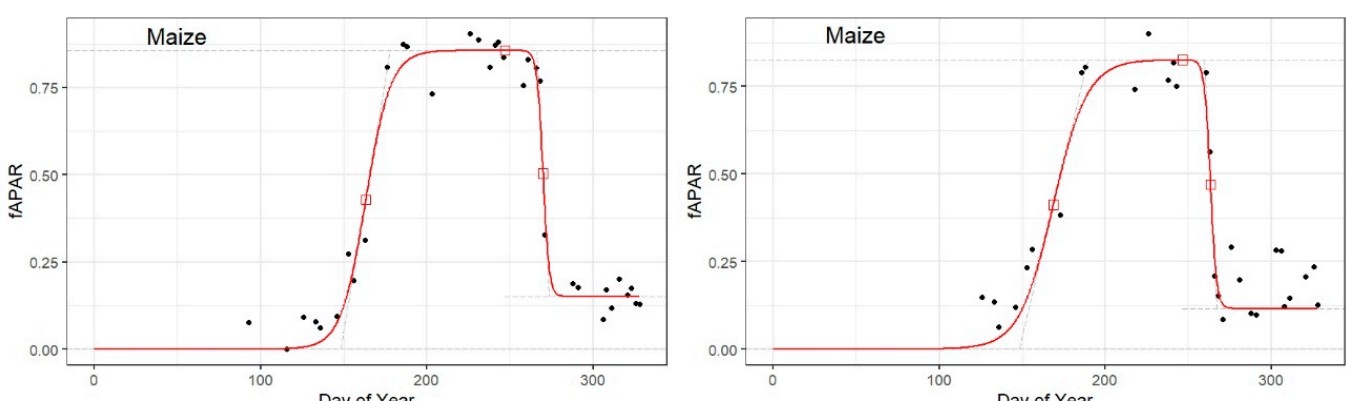

**Figure 12.** Relationship between Sentinel-2-, DMC-, and DHP-derived fCover during three years of measurement in arable fields. Each dot represents the median value for a field. The straight line represents the 1:1 line.

**Figure 13.** *Cont*.

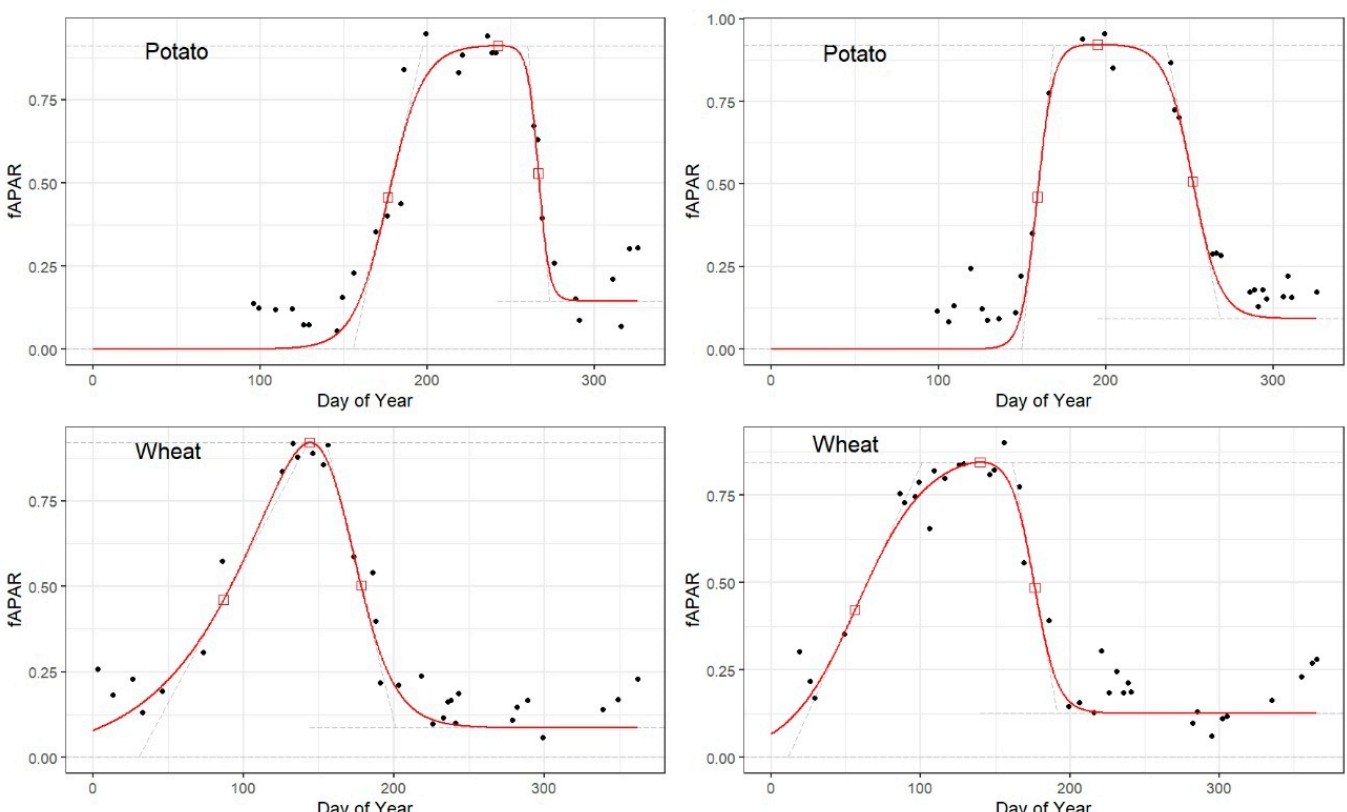

**Figure 13.** Examples of model fit of fAPAR for two maize, potato, and wheat fields during the 2017 season. The red squares depict the inflection points.

For the three arable crops, the first inflection point (midpoint 1, Figure 14) was characterized by low Sentinel-2-derived fAPAR values at the onset of the crop growing season and corresponded predominantly to the phenological stage of stem elongation, reflecting a point of maximum growth and subsequent increase in fAPAR (Figure 14). The endpoint of the upward sigmoid model marked the stage of canopy closure and extended to the stage of inflorescence emergence. Maximum fAPAR values (Max Point) were associated with flowering and fruit development, whereby the phenological stages of canopy closure, inflorescence emergence, and fruit development follow each other very closely in time. The inflection point of the downward sigmoid model (Mid Point 2) corresponded with low fAPAR values towards the end of the season when fruit ripening and senescence take place.

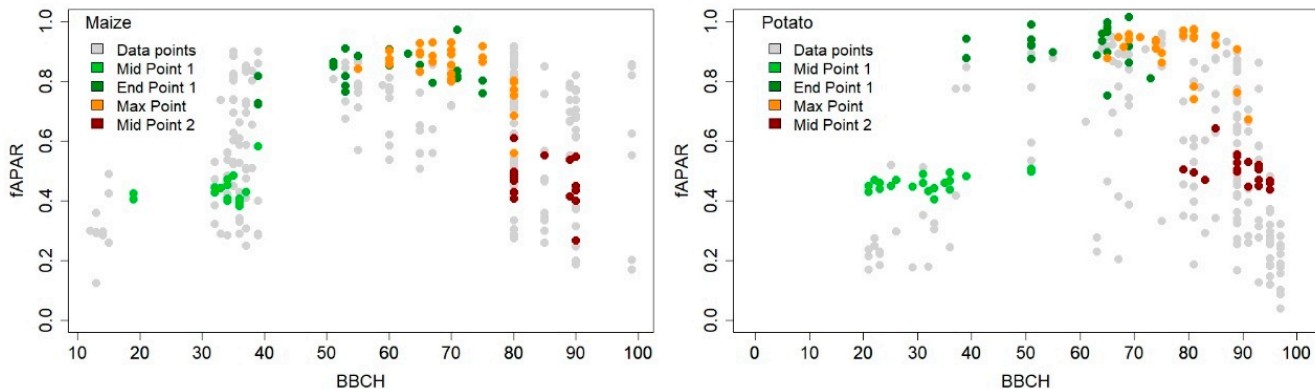

**Figure 14.** *Cont.*

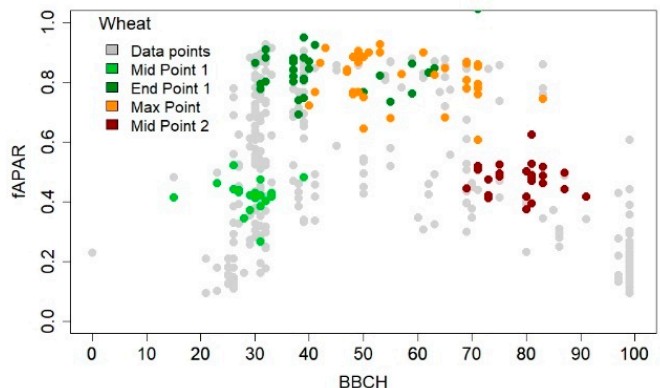

| BBCH | Potato | Maize | Wheat |
|------|--------|-------|-------|
| [10–20[ | Leaf Development | | |
| [20–30[ | Shoots | | Tillering |
| [30–40[ | Stem Elongation | | |
| [40–50[ | Tubers | | Booting |
| [50–60[ | Inflorescence Emergence | | |
| [60–70[ | Flowering | | |
| [70–80[ | Fruit Development | | |
| [80–90[ | Ripening | | |
| [90–99] | Senescence | | |

**Figure 14.** BBCH codes and mathematically derived model points of Sentinel-2 fAPAR for all maize, potato, and wheat fields during 2016–2017.

The DHP derived fAPAR values (Figure 15) displayed similar values and patterns as compared to Sentinel-2-derived fAPAR. Stem elongation (BBCH [30–40[, Figure 14) covered the largest range of fAPAR values and corresponded to vegetative crop growth.

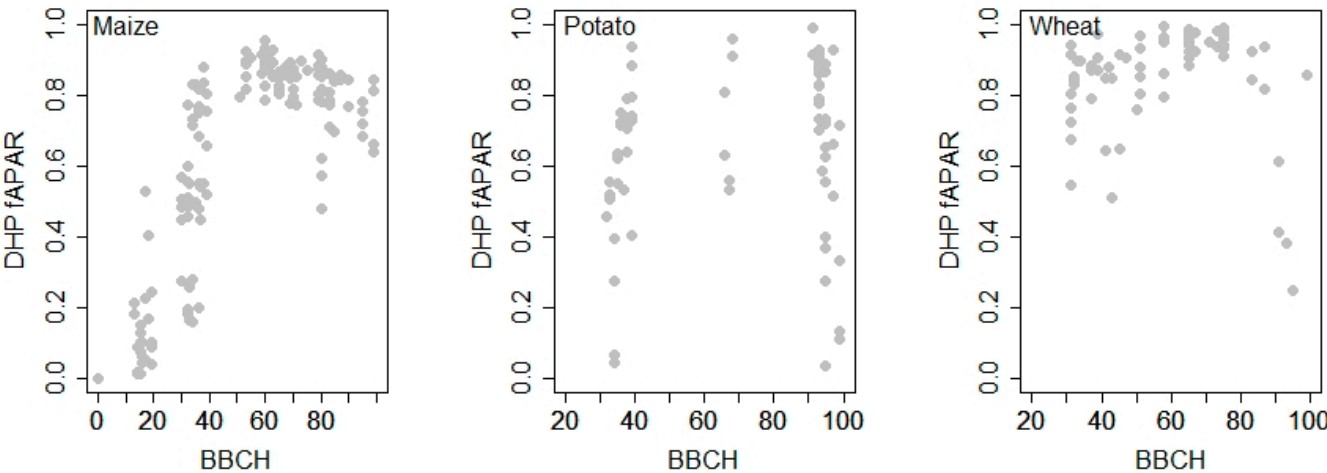

**Figure 15.** BBCH codes and DHP-derived fAPAR for maize, potato, and wheat for the period 2015–2017. See Figure 14 for BBCH codes.

## 4. Discussion

The demand for crop development stages, i.e., crop phenology, is high in an operational farming context, particularly in a changing climate with an increase in extreme weather events [55]. For farmers, the preferred unit of communication is days, to which field operations can be adjusted depending on the weather [56]. We reported crop development according to crop phenological stages as recorded by agronomists using the categorical BBCH scale [49], and we used calendar days in relation to the planting or sowing date. The BBCH observations and DHP acquisitions required regular field visits, certainly during the exponential growth phase, which was not always feasible in an operational context and did not allow a large number of agricultural parcels to be covered. Fixed pheno-cameras [57] or citizen science data acquisitions [58] may provide additional perspectives to augment in-situ observations. The disadvantages of fixed pheno-cameras and citizen science data acquisitions, such as their limited spatial coverage and observational bias, can be overcome by remote sensing.

Most remote sensing applications use time series smoothing algorithms to derive vegetation or crop development and relate this to yield [36,59]. We compared modeling methods such as cubic splines and sigmoid models to derive crop development. Cubic

splines required the definition of knots to fit cubic polynomials to the windows confined by the knots or cutpoints. However, the definition of cutpoints could only take place after the crop growing season and was therefore less suitable for an operational context within the growing season. The application of cubic splines in our case was limited to post-season assessments, as the models could only be successfully fitted after the harvest. Similar to our findings, sigmoid models have been successfully used to describe crop development throughout the growing season in maize [24,35] and winter wheat [25,34]. We have extended their use to within-season model fitting and included potato crop development.

A common presumption is that the requirements for field-scale monitoring are accomplished with the availability of global Sentinel-2 images at 10 to 20 m resolution every five days. The prerequisite to realizing improved crop monitoring through optical remote sensing data is that cloud-free images are available. Deep learning techniques have helped improve the detection of clouds and shadows and subsequently allowed the removal of contaminated observations prior to further time series analysis [60]. However, during periods of rapid change in the crop growing cycle, the number of observations may dwindle to a low number, necessitating a combination with other sensors such as UAV acquisitions [61], Sentinel-1-derived information [62], or high- and low-resolution data fusion [28]. Sentinel-1/2 integration is promising for parcel-based mapping and phenology detection [27,63]. In our study, we explored the use of models for single-sensor-derived biophysical variables such as fAPAR and fCover. fAPAR is integrated over the hemisphere to capture photosynthetic activity, whereas fCover is a near-nadir observation to obtain the fraction of soil covered by green vegetation.

Our results using satellite sensor-derived biophysical variables demonstrated that fCover displayed less variability than fAPAR during the onset of vegetative growth, while fAPAR showed less variability from vegetative growth to crop maturity stages. While satellite sensor-derived fCover is successful in detecting green-up and canopy closure, fAPAR provides information on the crop's photosynthetic activity [42,47]. Therefore, we preferred fAPAR to infer crop phenology and used sigmoid models to describe crop phenological development. Sigmoid models have the advantage that they can be fitted within the season when the vegetation indicator values are close to their maximum. Prior to this point, models can be fitted with less certainty, for example, by incorporating crop establishment and exponential vegetative growth. Both DHP and satellite sensor-derived biophysical variables may occur in the saturated zone at their highest values when the crop canopy is dense [50,64]. The difference between satellite-derived and DHP-derived biophysical variables can be explained by the differences in observation frequency, sampling, and calculation methods. While DHP images were processed using fisheye distortion correction followed by can-eye software, satellite images were processed using a biophysical variable and neural network (Figure 1).

The crop phenological stages were related to fitted sigmoid fAPAR models, whereby the inflection points indicated clear changes in the time series commensurate with changes in the crop status. Crop phenological stages such as stem elongation, full cover, and senescence were successfully detected. Certain phenological stages, such as inflorescence emergence or flowering in cereals, are short in duration and therefore more difficult to detect. According to the experimental results, the sowing stage appears to be more difficult to detect. Similarly, crop emergence detection, though not the focus of this research, proves challenging and requires multiple sensors at a high spatiotemporal resolution to arrive at a 6 to 10-day accuracy [31]. Moreover, sowing and planting dates, when not recorded, are important for crop modeling of current seasons and future scenarios [65,66]. Short duration phenological stages could therefore benefit from higher spatiotemporal and spectral information. Information about variability during the cropping season helps decide which biophysical variable to use for detecting crop phenological stages.

Variability between fields and years has an impact on the crop phenological progression and depends on soil moisture, temperature, and light incidence, as established in model inter-model comparisons for maize, wheat, and potato [16,17,21], which simulated

crop phenology well. Our methodology was tested on these three arable crops, which are common in Belgium and can be applied to other crops and regions of the world. A further comparison between crop phenological models and satellite sensor-derived biophysical variables could benefit the advanced understanding of weather impacts on phenological development and the subsequent scope of monitoring using remote sensing-derived biophysical variables.

## 5. Conclusions

The biophysical variables fraction of absorbed photosynthetically active radiation (fAPAR) and vegetation cover fraction (fCover) derived from digital hemispherical photographs (DHP), the disaster monitoring constellation (DMC), and Sentinel-2 (S2) satellite imagery were sensor-independent but crop-specific when monitoring the phenological stages of maize, wheat, and potato in Belgium. fCover showed less variability at the beginning of the crop-growing season, whereas fAPAR displayed less variability during vegetative development and mature crop growth. Since the dynamics of phenological development were well captured by fAPAR, a high goodness of fit was obtained. A numerical solution to fit sigmoid models through time series of biophysical variables enabled the derivation of inflection points and the subsequent detection of changes in green vegetation status and growth rate. In addition, the upward-sigmoid model demonstrated the ability to infer within-season phenological stages of crops, enabling crop performance assessment and within-season crop management decisions. Higher spatiotemporal sensor data and additional spectral information would assist in the detection of crop phenological stages, particularly short-duration stages, to support improved crop performance and yield modeling.

**Author Contributions:** Conceptualization, A.G.; methodology, A.G., Y.C., C.D., P.D., J.W., J.-P.G. and M.W.; software, A.G. and M.W.; validation, A.G., Y.C., J.W., C.D., P.D. and J.-P.G.; formal analysis, A.G.; investigation, A.G., A.-H.M.S., Y.C., J.W. and C.D.; data curation, A.G., A.-H.M.S., Y.C., C.D. and I.P.; writing—original draft preparation, A.G.; writing—review and editing, A.G., B.T., M.W., Y.C., J.-P.G. and P.D.; visualization, A.G.; funding acquisition, P.D., V.P., J.-P.G., A.G., I.P. and B.T. All authors have read and agreed to the published version of the manuscript.

**Funding:** The research was funded by the Belgian Science Policy Office (BELSPO), grant numbers SR/00/300 (BELCAM) and SR/67/392 (PHENOBEL).

**Data Availability Statement:** Data can be made available upon request and in agreement with the data owners.

**Acknowledgments:** We are grateful to the Pilot Centres for Arable Crops in Belgium, who contributed to the project with frequent field visits. We acknowledge the contribution of job student Brecht Bamps, who cross-checked the archive of photographs for phenological classification.

**Conflicts of Interest:** The authors declare no conflict of interest.

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
