# Peer review of "Crop Phenology Modelling Using Proximal and Satellite Sensor Data"

_remotesensing, doi:10.3390/rs15082090_

Round 1

Reviewer 1 Report

The manuscript regards the development of a model to infer phenology of some important crops in Belgium using ground (hemispherical photographs) and satellite optical data.

The work is written in good English.

Unfortunately the manuscript is affected by some majour issues such as:

- The ABSTRACT is chaotic and poorly written. The section from "We hypothesized that satellites (line 18)" to "crop phenological stages (line 23)" should be placed at the end of the abstract. Also a brief description of the materials and methods is missing.
- INRODUCTION: a short introduction must be written specifying that numerous models have been developed that simulate phenology. This part should be placed on line 56 before the section "All 27 crop models included in [16]..."
- Line 163: "Were the camera and fish-eye lens calibrated to establish coordinates?" How can the calibration of an optical lens derive coordinates?
- Figure 12: I don't understand the difference between the graphs of the right column and those of the left.
- DISCUSSION: The first (big) section, from "The demand for crop development stages..." up to line 366 "variables such as fAPAR and fCover", needs to be moved to the introduction.

I recommend a major review to the authors.

Author Response

Dear Reviewer,

Best wishes, on behalf of the authors,

Anne Gobin

Reviewer 2 Report

I congratulate the authors for carrying out this research and point out my general considerations below.

The research presents a way of monitoring the phenology of some crops through remote sensing techniques. The results provide essential information for the field of crop monitoring. Significant relationships are presented through scientific evidence but still need to be improved in an organization, making the text a little complicated.

After reading and correcting point by point in parts of the text, the publication of this research in the remote sensing journal can contribute to the areas of agricultural monitoring. Therefore, I recommend publication after the expressive correction of several points highlighted in the attached file.

Author Response

(The authors gave the same response as above.)
